# Statistical mechanics of exponentially many low lying states

**Swapnamay Mondal**

Department of Physics, Institute of Science, Banaras Hindu University,
Varanasi, 221005, India
Dublin Institute for Advanced Studies, 10 Burlington Road, Dublin, Ireland
School of Mathematics, Trinity College, Dublin 2, Ireland

swapno@maths.tcd.ie

## Abstract

It has recently been argued that for near-extremal black holes, the entropy and the energy above extremality respectively receive a $log\, T$ and a $T$-linear correction, where $T$ is the temperature. We show that both these features can be derived in a "low but not too low" temperature regime, by assuming the existence of exponentially many low lying states cleanly separated from rest of the spectrum, without using any specific theory. Argument of the logarithm in the expression of entropy is seen to be the ratio of temperature and the bandwidth of the low lying states. We argue that such spectrum might arise in non-supersymmetric extremal brane systems. Our findings strengthen Page's suggestion that there is no true degeneracy for non-supersymmetric extremal black holes.

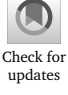

# 1 Introduction

Microscopic explanation of the entropy of non-extremal black holes has received a fair degree of attention [1–16]. However, due to non-zero temperature, there is more to the thermodynamics of such black holes than just entropy. E.g. semi-classical analysis entails that slightly above extremality, energy or mass of such black holes grows quadratically with temperature. The proportionality constant, which has the dimension of inverse mass, gives a mass scale $M_{gap}$. It has been suggested that below this scale, Hawking's analysis may not be trustable [17–19]. Microscopic descriptions, such as [20, 21] indeed exhibit a gapped energy spectrum. But these descriptions are limited to supersymmetric black holes.

Recently there has been a surge of interest in near-extremal black holes [22–28]. In particular, careful consideration of quantum effects suggests the absence of a gap for near-extremal Reissner-Nordström black holes [29]. This comes from certain boundary modes, captured by the Schwarzian theory [30–44]. This sector changes the low temperature thermodynamics significantly, in particular giving rise to a $log\,T$ dependence in entropy and T-linear term in energy, where $T$ is the temperature. This in particular entails a negative infinite entropy at zero temperature. This feature however is inconsistent with a discreet spectrum, which a black hole is expected to have. Therefore the above mentioned corrections most likely are approximations to the true situation and begs the question - *what sort of discreet spectra can lead to such low temperature thermodynamics?* In fact, there is good reason to suspect that these features might not be exclusive to the Schwarzian theory, but to a class of theories comprising the Schwarzian theory. If so, what defines this class?

To begin with, the reason for this suspicion is two fold.

1. Firstly, given the considerable success of D-brane descriptions in reproducing the entropy of near-extremal black holes from microscopic state counting, it is conceivable that in the D-brane description some further detail of the low lying spectra might survive as well. The Schwarzian theory itself is unlikely to survive though. For near-extremal black holes, it arises as a consequence of the symmetry of nAdS$_2$ spaces and hence is specific to the gravity side. It is far from clear that such symmetries would arise in the corresponding D-brane set up. Emergence of Schwarzian theory[1] in the context of SYK model(s) may be a ray of hope, but SYK model or any cousin thereof has not been derived in a D-brane set up to the best of our knowledge. So a reasonable expectation would be that some qualitative features of the low lying spectra, such as low temperature thermodynamics might survive in the D-brane description.[2]

2. Secondly, whereas the connection between strange metals and black holes (e.g. similarity of Planckian time scale in strange metals [46, 47] and ringdown time scale in black holes [48, 49] ) has long been noted, it is not entirely clear whether these similarities emerge from some microscopic similarity. Application of holography in such contexts indirectly suggest absence of quasi particles might be a rather consequential microscopic similarity. Existence of the SYK model [30–32, 50], which captures key features of both sides further emboldens this possibility.

Thus there is good reason to ask- *can one understand $log\,T$ dependence of entropy and T-linear energy, as consequences of some broad features, without using any specific theory?* We suggest that the presence of exponentially many low lying states, or equivalently absence of quasi-particles might be a necessary condition to realize such low temperature thermodynamics.

---

[1]In this paper, we have non-supersymmetric Schwarzian theory in mind. There are key differences with supersymmetric versions thereof.

[2]For conformal field theory descriptions, it has been suggested that $T\bar{T}$ deformation captures such a change [45].

We set out by noting that for non-supersymmetric extremal black holes, a key question is whether the ground state is degenerate or not. Extremality suggests a degenerate ground state, whereas lack of supersymmetry suggests a non-degenerate one. Page had previously argued in favor of a non-degenerate spectrum [19], and so has [29]. This question is relevant for the microscopic D-brane description as well and in this case, we advocate a specific case of Page's suggestion, namely exponentially many low lying states, spread over an energy window $\Delta$, cleanly separated from rest of the spectrum by a gap $E_{gap}$ (not to be confused with $M_{gap}$). We argue such spectra are likely to arise from D-brane constructions and analyze the statistical mechanics of such systems under the assumption of genericness in different low temperature ($T \ll E_{gap}$) regimes. We find that in a *low but not too low* temperature regime, qualitative features of the Schwarzian thermodynamics, such as $log\, T$ dependence in entropy and $T$-linear dependence of energy, are readily retrieved. In particular $\Delta$ is seen to play the role of $M_{gap}$. This suggests that the presence of exponentially many low lying states might be a robust feature for non-supersymmetric near extremal systems and possibly an essential feature to realize $log\, T$ dependence in entropy and a $T$-linear energy at low temperatures.

The rest of the paper is organized as follows. In section 2, we survey well known systems with exponentially many low lying states. In section 3, we first argue that extremal non-supersymmetric brane systems might exhibit spectra with exponentially many low lying states cleanly separated from rest of the spectrum by a gap. Then we argue that generically the low lying band of states is well approximated by a band of equi-spaced states. Lastly we analyze the statistical mechanics of such a system and show they capture the qualitative features of the Schwarzian thermodynamics for "low but not too low" temperatures. In section 4, we discuss implications of our findings.

## 2 Systems with large number of low lying states

In this section we quickly review some physical systems, which exhibit or are likely to exhibit exponentially many low lying states.

**Near-extremal black holes:** Extremal black holes carry zero temperature and hence their entropy is the logarithm of the ground state degeneracy. It is unclear though what protects this degeneracy. It has been argued that exponential of the extremal entropy actually counts not only the ground states but all the states below a certain gap scale [19]. Anyhow, technically extremal black holes are easier to deal with owing to the decoupling of the near-horizon region.

For non-extremal black holes, the near horizon region does not decouple. It is useful to separate such a spacetime into a Near Horizon Region (NHR) and a Far Away Region (FAR) [28]. For the near extremal case, the NHR region still retains some flavours of the $AdS_2 \times S^2$ space. After reduction over transverse $S^2$, one gets an effective theory on the two dimensional space spanned by the radial and temporal coordinates. This theory in particular incorporates Jackiw Teitelboim (JT) gravity [51, 52], which appears abundantly in the context of near-extremal black holes [53–63].

Both the $AdS_2$ and $S^2$ receive temperature corrections of same order, leading to the breaking of $AdS_2$ isometries in the NHR. This symmetry breaking scale has been identified with the gap scale $M_{gap}$ [54]. The low temperature dynamics is governed by the (pseudo)Goldstones of this symmetry breaking [33]. For more on Kaluza Klein reduction in this settings, see [64,65]. After careful consideration of one loop correction of the JT mode, one obtains the following

expression for entropy and energy (above extremality) [29]

$$S = S_0 + \frac{4\pi^2 \Phi_{b,Q}}{\beta} - \frac{3}{2} \log \frac{\beta}{3\Phi_{b,Q}} \,,$$

$$E = \frac{2\pi^2 \Phi_{b,Q}}{\beta^2} + \frac{3}{2\beta} \,, \tag{1}$$

where $\Phi_{b,Q} \sim M_{gap}^{-1}$ is the value of the dilaton at fixed charge $Q$ and $\beta$ is the inverse temperature. For very low temperatures, energy above extremality scales linearly with temperature, thus lacking any characteristic energy scale and hence suggesting a gapless spectrum. Since this follows from a gravity analysis, rather than a microscopic one, the gapless spectrum is better thought of as a smeared version of densely spaced low lying states. It might seem at odds with vanishing density of states near the ground state, as suggested by $\lim_{T \to 0} e^S = 0$. However there is no contradiction, as the "small" density of states can still be large in system size. In fact, this possibility is explicitly realized for SYK model, as we shall see later in this section.

**Brane systems:**   Microscopic descriptions of black holes are often constructed in terms of an effective string, wrapping a circle of size $R$ [66,67]. For example, for D4 brane black holes in $\mathcal{N} = 2$ String theory, this string represents an M5 brane reduced over a four-cycle inside the Calabi-Yau threeefold, and wraps the M-theory circle. Charges carried by the black hole appear as the conserved charges in the worldvolume theory on this effective string. Hence black hole microstate counting amounts to counting states in the worldvolume theory in a given charge sector. To facilitate this task, one goes to a regime, where the system is well described by a free conformal field theory (CFT), where central charge can be determined solely from the field content. Then one uses Cardy's formula to count the states [68]. Such a CFT captures a family of black holes at one go, including non-extremal ones. In fact near-extremal black hole entropies have been successfully reproduced from such state counting in a number of cases [3–16].

Note that it is not that one writes down a worldvolume Lagrangian and it happens to describe a CFT (as is the case for D3 branes). Rather, one simply looks at the massless field content and assumes that in an appropriate regime their interaction can be ignored, which is sometimes justified by dilute nature of the system at low energies [3]. In some instances, a sigma model description of the effective string dynamics also exists [69].

Anyhow, the worldvolume theory has no reason to be conformal to start with and the CFT description is valid only in a particular corner of the moduli space.[3] Away from the conformal point, one expects the degeneracy to be protected for the supersymmetric states, but not for the non-supersymmetric ones. The degenerate non-supersymmetric states of the CFT are likely to split, as one moves away from the CFT point. For near-supersymmetric states, the splitting is likely to be small, leading to large number of closely spaced states.

Such near-supersymmetric states can sometimes be associated with a brane-antibrane system [5, 9, 11]. Should one consider the low energy theory describing the specific brane-antibrane system, it is conceivable that the near-extremal states would appear as low energy states in this theory.

A particularly curious case is that of extremal non-supersymmetric states. Since these are not even nearly supersymmetric, corresponding states need not remain even nearly degenerate as one moves away from the CFT point.[4] On the other hand semi-classical gravity would have

---

[3]For D4-brane black holes, this CFT limit point can be rephrased as a decoupling limit [45,70].

[4]Note that the moving away from CFT point discussed here is not of the sort discussed in [45], which captures gravitational effects and does not affect the degeneracies.

us believe that this state is highly degenerate. To resolve this tension a study the quantum mechanics of such systems is required. We will revisit this issue in section (3.1).

**Sachdev-Ye-Kitaev model:** SYK model [30–32, 50] is a model of disordered Majorana fermions. It's claims to fame are manifold. It is solvable in a specific large $N$ limit and saturates the chaos bound[5] [73]. In the same regime, it develops an emergent reparameterization symmetry, which is both spontaneously and explicitly broken, resulting in (pseudo)Goldstones, which dominate the low energy physics. This pattern of symmetry breaking is reminiscent of near $AdS_2$ spaces [33].

One key feature of SYK model is it's lack of quasiparticles. Such systems differ from systems with quasi particles for the following reasons. The energies of quasi particles depend on momentum as power law, and the smallest non-zero momentum goes inversely with the system size, together implying that systems with quasi particles have polynomially small gaps in system size, or equivalently polynomially many low lying states. In particular this entails vanishing entropy density at low temperatures in thermodynamic limit.

On the other hand, if a system exhibits exponentially small level spacing in system size, then a small energy interval above the ground state contains exponentially many states leading to non-zero entropy density at low temperatures. However for temperatures exponentially small in system size, an energy window would contain few states, leading to vanishing density of states or negative infinite entropy (as in (1)). Systems with exponentially many low lying states are necessarily systems without quasi-particles, such as the SYK model. It is worth quickly recollecting how the above mentioned thermodynamic features are realized in the SYK model.

SYK model is solved in a $N \to \infty$, $\beta J \to \infty$ limit, where $N$ is the number of fermions, $\beta$ is the inverse temperature and $J$ is the coupling strength. In this limit, there is an emergent reparameterization symmetry, which is spontaneously broken down to $SL(2,R)$ by the mean field solution. The presence of this symmetry is a direct indication of lack of quasi-particles. In this limit the system exhibits a non-zero entropy density at zero temperature [32]. This does not quite represent a degenerate ground state though. Certain pathologies in the four point function forces one to move away from the above mentioned limit and take $1/\beta J$ corrections into account and thereby explicitly break the emergent reparameterization symmetry. This also means that the above mentioned mean field solutions and results obtained therefrom can not be pushed all the way to zero temperature. These include the non-zero entropy density at zero temperature. Such would be the situation for exponentially many low lying states, as explained in the last paragraph. Altogether this means the results obtained in $N \to \infty, \beta J \to \infty$, are really valid in a "low but not too low" temperature regime. This temperature regime will reappear in coming sections.

This can be rephrased as the non-commutativity of large system size limit and zero temperature limit. If one takes the large system size limit first, then arbitrarily small energy window contains exponentially many states, leading to a non-vanishing zero temperature entropy [32]. The other order of limits gives vanishing density of states.

This is succinctly captured by the density of states [35, 74–76]

$$D(E) \sim \frac{1}{N} e^{N s_0} \sinh \sqrt{2N\gamma E} \,, \tag{2}$$

where the constants $s_0, \gamma$ are not particularly relevant for us. The factor $e^{N s_0}$ ensures the existence of an extensive entropy. Note, the degeneracy is exponentially large in system size and at the same time it vanishes at low energies, just like near-extremal black holes discussed below (1).

---

[5]The bound on chaos has been related to Einstein velocity bound [71, 72].

# 3 Statistical mechanics of exponentially many low lying states

What kind of spectra can explain the thermodynamics of near-extremal black holes? This question has been asked before, e.g. in [19], where it was suggested that a non-supersymmetric extremal black hole is likely to have no degeneracies, but only states separated by exponentially small energy gaps. This argument was based on semi-classical black hole thermodynamics. Lately, it has been realized that quantum effects significantly alter the semi-classical black hole thermodynamics at low temperatures [29]. At high temperatures, the semi-classical thermodynamics and hence the analysis of [19] are still valid. But for low temperatures, we must ask afresh - *what kind of spectra can explain the low temperature thermodynamics of near-extremal black holes?*

One could proceed by extracting the density of states from the partition function using Laplace transform. But this approach really gives an averaged out picture, not the true discrete spectrum. Moreover, it does not give much insight about the class of spectra that might exhibit similar low temperature thermodynamics. We take the converse approach, i.e. first we make an educated guess about the low lying spectra and then check if it reproduces the qualitative behaviour of near-extremal thermodynamics at low temperatures, i.e. the $\log T$ piece in entropy and linear in $T$ piece in energy above extremality.

To do so, we note that a near-extremal black hole is likely described as a thermal excitation of the corresponding extremal object.[6] So the low lying spectrum of the extremal object is the key to the low temperature thermodynamic properties. More specifically we consider extremal non-supersymmetric systems. For such systems, extremality still predicts a highly degenerate ground state, whereas the absence of supersymmetry suggests a non-degenerate ground state. Between these two extremes, a conceivable middle way could be a band of low lying nearly degenerate states, cleanly separated from rest of the spectrum and accounting for the zero temperature entropy.

To verify this, one would ideally need an explicit microscopic description of such brane systems, which is beyond the scope of this paper. However we note that given a supersymmetric brane system with multiple charges, one can simply alter various charges to get extremal non-supersymmetric black holes [77]. Since the building blocks of the system are still supersymmetric, one expects the low energy theory of this brane system to be a non-supersymmetric theory, yet built out of superfields. In the following section, we provide evidence that such systems can indeed host large number of low lying states with energies parametrically smaller compared to other excited states, thus realizing the above mentioned situation.

## 3.1 Systems with exponentially many low lying states below a gap

For specificity, we consider 4-charge black holes in $\mathcal{N} = 8$ string theory in four spacetime dimensions, in a duality frame where all charges are Ramond Ramond. Corresponding microscopic description comprises 4 stacks of D-branes, wrapping different cycles of the internal six-torus [20, 21]. Any three stacks collectively preserve 1/8th of the supersymmetry, i.e 4 supercharges. Depending on orientation, the remaining stack may or may not preserve these supercharges. Thus depending on relative orientation of stacks, we get either a supersymmetric or a non-supersymmetric black hole. However, this difference is undetectable at the level of any two stacks or even three stacks. E.g. denoting the collection of $i^{th}$, $j^{th}$ and $k^{th}$ stack as $(ijk)$, the triplets of stacks (123) or (124) would both preserve 4 supercharges and would not know whether the full system is supersymmetric or not. In the supersymmetric case both triplets would preserve the same set of 4 supercharges, whereas in the non-supersymmetric

---

[6]Brane-antibrane systems have successfully reproduced the non-extremal entropy [2–16]. We are not aware of a microscopic understanding of the temperature in these settings though.

case they would preserve different sets of 4 supercharges. These different sets of 4 supercharges can be thought of as different $\mathcal{N} = 1$ subalgebras of $\mathcal{N} = 2$ supersymmetry (in four dimensional language) preserved by the pair of stacks (12), which is common to both (123) and (124). These two $\mathcal{N} = 1$ subalgebras are thus related by $SU(2)_R$ R-symmetry rotations of the $\mathcal{N} = 2$ supersymmetry. Hence fields associated with the pair (12) will appear both in their "original" as well as R-symmetry rotated forms in the Lagrangian describing the low energy dynamics of the brane system. Note that for $\mathcal{N} = 1$ chiral multiplet scalars inside the $\mathcal{N} = 2$ hypermultiplet, such rotation involves taking hermitian conjugate. Although this is a symmetry for the pair (12), for the full brane system this compromises supersymmetry.

A detailed analysis of the above mentioned system is beyond the scope of this paper. But we do note that the non-supersymmetric worldvolume theory is of a very specific sort, namely made of superfields. We now argue that such systems are likely to exhibit a particular type of spectrum, namely when they have degenerate classical vacua, the quantum spectrum has band of low lying states with parametrically smaller energies compared to the excited states.

We start with a system involving a single chiral superfield (in four dimensional language), reduced to one dimension. A chiral superfield $\Phi = (\phi, \psi, F)$ can be expanded as

$$\Phi = \phi + i\theta\sigma^\mu\bar{\theta}\partial_\mu\phi - \frac{1}{4}\theta^2\bar{\theta}^2\Box\phi + \sqrt{2}\theta\psi - \frac{i}{\sqrt{2}}\theta^2\partial_\mu\psi\sigma^\mu\bar{\theta} + \theta^2 F, \tag{3}$$

$$\Phi^\dagger = \phi^\dagger - i\theta\sigma^\mu\bar{\theta}\partial_\mu\phi^\dagger - \frac{1}{4}\theta^2\bar{\theta}^2\Box\phi^\dagger + \sqrt{2}\bar{\theta}\bar{\psi} + \frac{i}{\sqrt{2}}\bar{\theta}^2\theta\sigma^\mu\partial_\mu\bar{\psi} + \bar{\theta}^2 F^\dagger. \tag{4}$$

Consider the Lagrangian

$$\mathcal{L} = \int d^4\theta\, \Phi^\dagger\Phi + \int d^2\theta\, W(\Phi) + \int d^2\bar{\theta}\, \overline{W}(\Phi^\dagger)$$

$$= \partial_\mu\phi^\dagger\partial^\mu\phi + F^\dagger F - i\bar{\psi}\bar{\sigma}^\mu\partial_\mu\psi + \frac{\partial W}{\partial\phi}F + \frac{\partial\overline{W}}{\partial\phi^\dagger}F^\dagger - \frac{1}{2}\frac{\partial^2 W}{\partial\phi^2}\psi^2 - \frac{1}{2}\frac{\partial^2\overline{W}}{\partial(\phi^\dagger)^2}\bar{\psi}^2. \tag{5}$$

Now define $\tilde{\Phi} := (\phi^\dagger, \psi, F)$. If $\Phi$ is a $\mathcal{N} = 1$ chiral multiplet in a $\mathcal{N} = 2$ hypermultiplet, then $\tilde{\Phi}$ is the R-symmetry rotated form of $\Phi$. Thus a Lagrangian fashioning both superfields, will break supersymmetry altogether.

In order to consider such a Lagrangian, we first note that the kinetic term gives same terms for either superfields, whereas the superpotential terms make a distinction. On top of the Lagrangian (5), let us add an extra superpotential term

$$\int d^2\theta\, W'(\tilde{\Phi}) + \int d^2\bar{\theta}\, \overline{W'}(\tilde{\Phi}^\dagger) = \frac{\partial W'}{\partial\phi^\dagger}F + \frac{\partial\overline{W'}}{\partial\phi}F^\dagger - \frac{1}{2}\frac{\partial^2 W'}{\partial(\phi^\dagger)^2}\psi^2 - \frac{1}{2}\frac{\partial^2\overline{W'}}{\partial(\phi)^2}\bar{\psi}^2. \tag{6}$$

This leads to the total F-term potential

$$V_F = \left|\frac{\partial W}{\partial\phi} + \frac{\partial W'}{\partial\phi^\dagger}\right|^2. \tag{7}$$

Let $\phi_0$ be a global minima of the potential $V_F$, i.e. $\left(\frac{\partial W}{\partial\phi} + \frac{\partial W'}{\partial\phi^\dagger}\right)\Big|_{\phi_0} = 0$.[7] Let us expand around the minima $\phi = \phi_0 + \varphi$. Then up to quadratic order in $\varphi$, the potential around $\phi_0$ is

$$V_F = \left|M + M'\right|^2 |\varphi|^2 + \ldots, \tag{8}$$

---

[7]Note that for multiple chiral fields it is not obvious that the F-term equations can be satisfied.

where

$$M = \frac{\partial^2 W}{\partial \phi^2}\Big|_{\phi_0}, \qquad M' = \frac{\partial^2 W'}{\partial(\phi^\dagger)^2}\Big|_{\phi_0}. \tag{9}$$

Defining $\varphi = \frac{1}{\sqrt{2}}(x + iy)$, we see the bosonic ground state energy to be

$$E_0^B = \hbar\left|M + M'\right|. \tag{10}$$

The fermionic Hamiltonian

$$H_F = -(M + M')\psi^1\psi^2 - (M + M')^*\bar{\psi}^1\bar{\psi}^2, \tag{11}$$

has the fermionic ground state energy

$$E_0^F = -\hbar\left|M + M'\right|. \tag{12}$$

Thus one has a vanishing ground state energy for the entire system, in quadratic approximation for the potential. However in absence of supersymmetry, the anharmonic terms will lead to non-vanishing ground state energy. The point is that such contributions start at $\mathcal{O}(\hbar^2)$, as we argue in the following.

First coming to the bosonic oscillator, the leading contribution of the $\varphi^3$ term occurs at second order in perturbation theory and is of $\mathcal{O}(\hbar)^2$. The leading contribution of the $\varphi^4$ term in potential appears in the first order in perturbation theory, with a $\mathcal{O}(\hbar)^2$ shift in energy. Thus the leading shift in the bosonic ground state energy is of $\mathcal{O}(\hbar)^2$.

As for the fermionic ground state energy, $M$ and $M'$ in (12) should be replaced by the vacuum expectation value of $\frac{\partial^2 W}{\partial \phi^2}$. The leading term is $\frac{\partial^2 W}{\partial \phi^2}\big|_{\phi_0}$. The linear in $\varphi$ term has vanishing vacuum expectation value, hence the first subleading term is quadratic in $\varphi$ term, which can be checked to have a $\mathcal{O}(\hbar)$ vaccum expectation value. Along with the overall factor of $\hbar$ in (12), this means that the corrections to $E_0^F$ also start at $\mathcal{O}(\hbar^2)$.

Altogether, the shift of ground state energy of the whole system is $\mathcal{O}(\hbar^2)$, as compared to the excited states, which have energy of $\mathcal{O}(\hbar)$. This means that the ground state has energy parametrically smaller than the lowest excited state.

If there are multiple global minima of $V_F$, by the same token, each minima will have its own "ground state" with an energy of $\mathcal{O}(\hbar^2)$. Altogether, one would get a "band of ground states"[8] with energies of $\mathcal{O}(\hbar^2)$ cleanly separated from the excited states, with energies of $\mathcal{O}(\hbar)$. A somewhat more pedestrian example of similar spectra is presented in appendix (A).

## 3.2 Generic systems with exponentially many low lying states below a gap

The exact distribution of the low lying states in the above example will depend on detail of the specific theory. In the absence of a specific theory, we can do no better than considering a generic spectrum. We will take such a spectrum to be made of states whose energy eigenvalues are chosen randomly from a given energy interval. For simplicity, we take the probability distribution of individual eigenvalues to be uniform. Then it can be shown that statistical average of spectra thus generated is the equi-spaced spectrum with the given number of states in the given interval. Next we provide some numerical evidence that for large number of states and not too low temperatures, the partition function of a randomly generated spectrum

---

[8]Degeneracy among these states is pretty unlikely. This is because every classical minima would generically have different anharmonic terms, hence different ground state energies. In case, some minima are related by some discreet symmetry, the corresponding ground states indeed will show degeneracy, only to be lifted by non-perturbative effects.

is indeed quite close to the partition function of the equi-spaced spectrum with same number of states distributed over same energy interval. This provides considerable motivation for studying the thermodynamics of the equi-spaced spectrum.

For the statistical analysis, we map a state with energy $E \in [0, \Delta]$ to a point $\frac{E}{\Delta} \in [0, 1]$. Thus the spectrum comprising $\Omega$ states in the interval $[0, 1]$ is mapped to a set of $\Omega$ points in the interval $[0, 1]$. Each point is chosen randomly from uniform probability distribution $p(x) = \theta(x)\theta(1 - x)$, where $\theta(x)$ is the Heaviside theta function. Now randomly draw $\Omega$ points. Let $x_i$ denote the position of the $i^{th}$ point. Here $1^{st}$ point stands for the leftmost point (i.e. the state with the lowest energy), $2^{nd}$ point stands for the second leftmost point, and so on. It can then be shown that the average separation of consecutive points $\overline{x_{i+1} - x_i} = 1/(\Omega + 1)$ for $i = 1, \ldots, \Omega - 1$ (see Appendix B for details). This means that the "average spectrum" is an equi-spaced one.

In the absence of enough information, average value of a variable is often taken to be the actual value of that variable. However it is not obvious how "close" a randomly chosen spectrum is to the equi-spaced one. We shall not attempt to answer this question rigorously. Instead, we do a few numerical experiments to check the simpler question of immediate relevance: whether the partition function of a randomly chosen spectrum is close to that of the equi-spaced spectrum? We find that for large $\Omega$ and not too low temperatures, indeed they are close.

To this end, we fix the energy interval to be $(0, 1)$ and consider different values of the number of states $\Omega$ distributed over this interval. Let $Z_{generic}$ be the partition function of a generic spectra, obtained by randomly choosing $\Omega$ energy levels in the interval $(0, 1)$ and let $Z_{equi-spaced}$ be the partition function of $\Omega$ evenly distributed states over the same interval.

In the following we plot $(Z_{generic}, Z_{equi-spaced})$ (in the top panel) and $\frac{Z_{generic} - Z_{equi-spaced}}{Z_{equi-spaced}}$ (in the bottom panel), for $\Omega = 10, 10^2, 10^3, 10^4$ and $10^5$. For each $\Omega$, we consider three instances of randomly generated spectra and compare the resultant partition functions $Z_{generic}$ (blue plots in top panels) with the partition functions $Z_{equi-spaced}$ of with equal number of evenly spaced states (orange plots in top panels).

For any $\Omega$, $Z_{generic}$ is seen to differ significantly for different realizations. For relatively small $\Omega$, $Z_{equi-spaced}$ differs considerably from any realization of $Z_{generic}$. But the differences diminish, both in absolute and relative terms for large $\Omega$.

### 3.3 Statistical mechanics of generic systems with exponentially many low lying states below a gap

Having argued that extremal non-supersymmetric brane systems are likely to have a low lying band of large number of states cleanly separated from rest of the spectrum and having shown that such a generic band is well approximated by uniformly distributed states, we now consider the idealized spectrum. Let the low lying uniform band contain $\Omega := e^{S_0}$ states, in an energy interval $(0, \Delta)$ and separated by a gap $E_{gap}$ from the rest of spectrum. In the following, we make the simplifying assumption $E_{gap} \gg \Delta$. As we will see $\Delta$ plays the role of $M_{gap}$ (1).

For temperatures $T \ll E_{gap}$, the contribution of states with $E \geq E_{gap}$ can be ignored in the partition function $Z$ and we can approximate

$$Z \simeq \sum_{E \leq \Delta} d(E)e^{-\beta E}, \tag{13}$$

where $d(E) = \sum_{n=1}^{\Omega} \delta(E - n\Delta/\Omega)$. With this approximation, the partition function can be written in closed form:

$$Z = \frac{1 - e^{-\Delta\beta}}{e^{\Delta\beta/\Omega} - 1}. \tag{14}$$

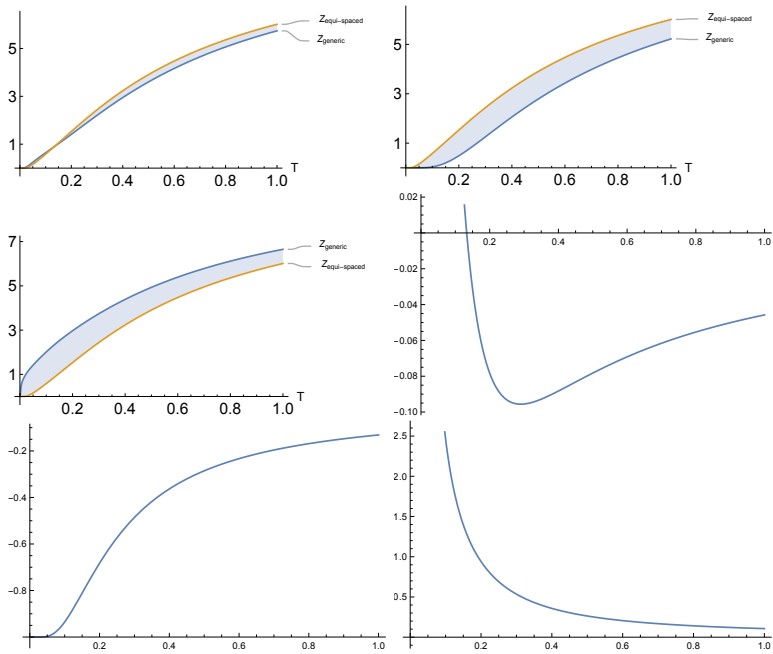

Figure 1: $(Z_{generic}, Z_{equi-spaced})$ versus temperature T (in the top panel) and $\frac{Z_{generic} - Z_{equi-spaced}}{Z_{equi-spaced}}$ versus temperature T (in the bottom panel), for three realizations with $\Omega = 10$.

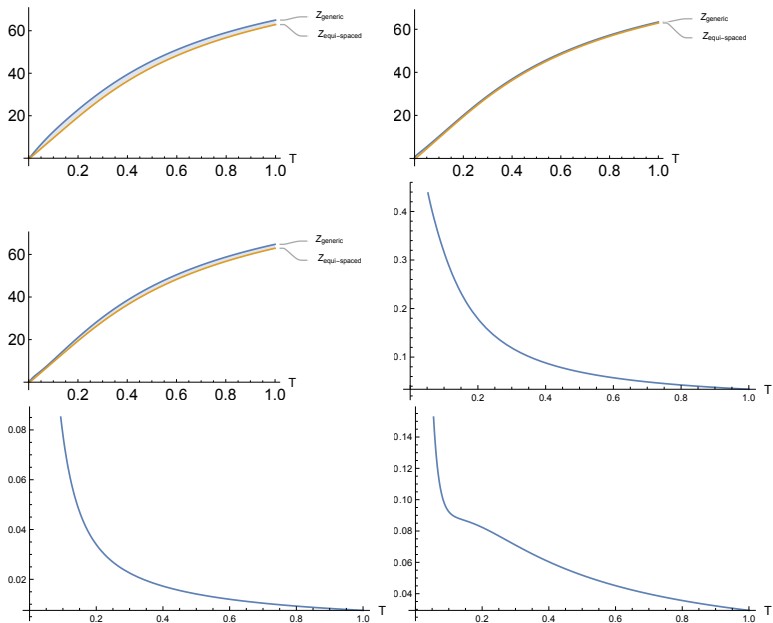

Figure 2: $(Z_{generic}, Z_{equi-spaced})$ versus temperature T (in the top panel) and $\frac{Z_{generic} - Z_{equi-spaced}}{Z_{equi-spaced}}$ versus temperature T (in the bottom panel), for three realizations with $\Omega = 100$.

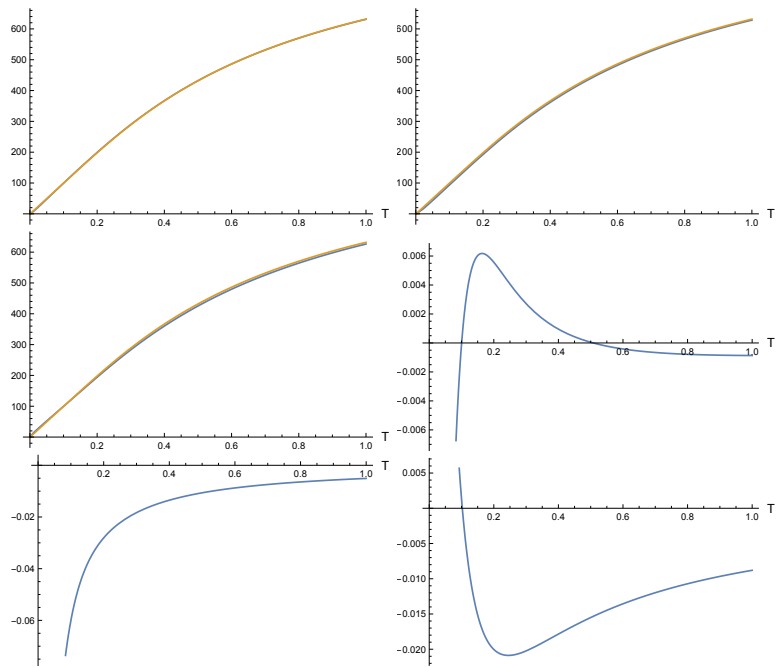

Figure 3: $(Z_{generic}, Z_{equi-spaced})$ (in the top panel) versus temperature T and $\frac{Z_{generic} - Z_{equi-spaced}}{Z_{equi-spaced}}$ versus temperature T (in the bottom panel), for three realizations with $\Omega = 1000$.

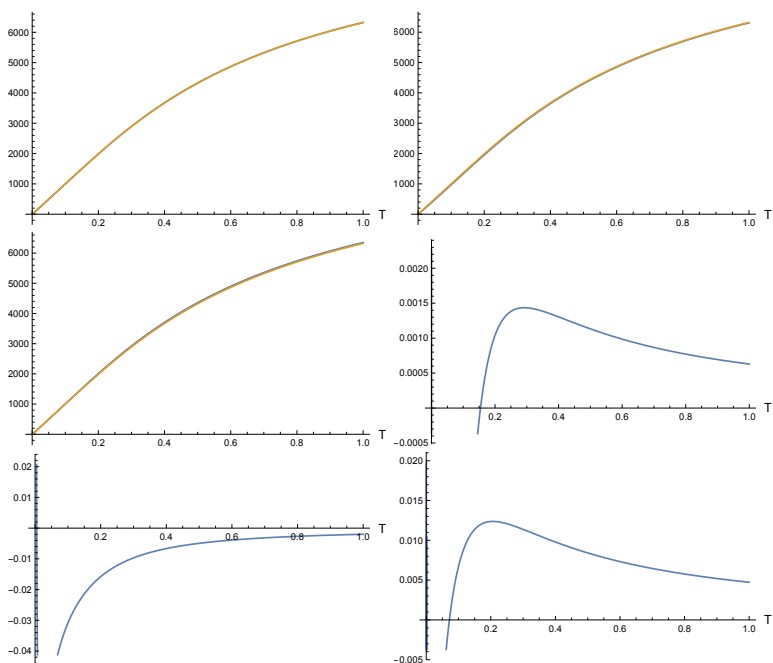

Figure 4: $(Z_{generic}, Z_{equi-spaced})$ versus temperature T (in the top panel) and $\frac{Z_{generic} - Z_{equi-spaced}}{Z_{equi-spaced}}$ versus temperature T (in the bottom panel), for three realizations with $\Omega = 10000$.

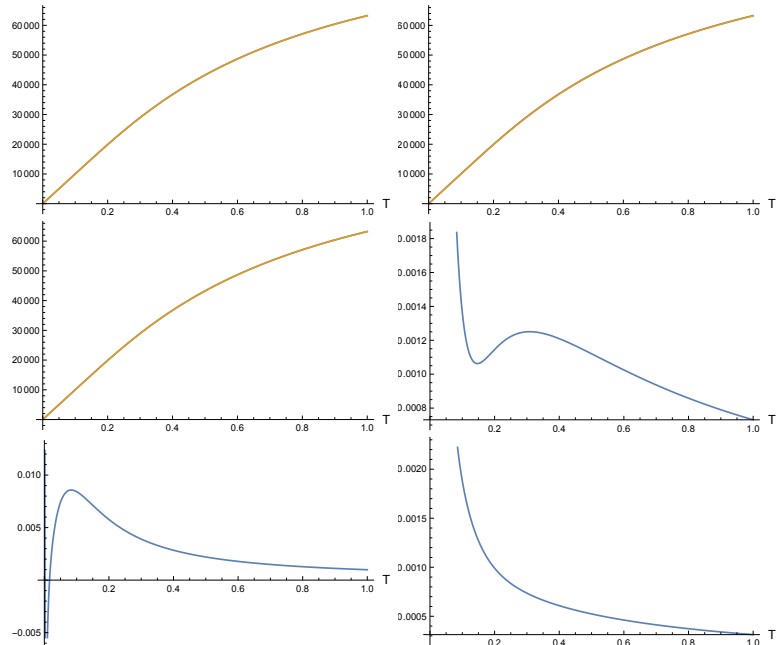

Figure 5: $(Z_{generic}, Z_{equi-spaced})$ versus temperature T (in the top panel) and $\frac{Z_{generic} - Z_{equi-spaced}}{Z_{equi-spaced}}$ versus temperature T (in the bottom panel), for three realizations with $\Omega = 100000$.

Since there are now two relevant energy scales, namely $\Delta$ and $\Delta/\Omega$, there are three low temperature regimes to be considered, as discussed in the following.

1. *Moderately low temperatures* $(E_{gap} \gg T \gg \Delta)$: In this regime, $\Delta\beta \ll 1$. Hence

$$\ln Z \simeq S_0 - \frac{1}{2}(1 + \Omega^{-1})\Delta\beta + \frac{1}{24}(1 - \Omega^{-2})\Delta^2\beta^2 + \dots,$$

$$U = -\partial_\beta \ln Z \simeq \frac{1}{2}(1 + 1/\Omega)\Delta - \frac{1}{12}(1 - 1/\Omega^2)\Delta^2\beta + \dots,$$

$$S = \beta U + \ln Z \simeq S_0 - \frac{1}{24}(1 - 1/\Omega^2)\Delta^2\beta^2 + \dots \tag{15}$$

We note (15) admits smooth $\Delta \to 0$ limit, as well as a "high temperature limit" $\Delta\beta \to 0$. Both have the entropy $S_0$, but the $\Delta\beta \to 0$ limit has a non-vanishing internal energy as well.

2. *Low but not too low temperatures* $(\Delta \gg T \gg \Delta/\Omega)$: In this regime, $\Delta\beta \gg 1$, but $\Delta\beta/\Omega \ll 1$. Hence

$$\ln Z \simeq S_0 - \ln\beta\Delta - \frac{\beta\Delta}{2\Omega} + \dots,$$

$$U \simeq T + \frac{\Delta}{2\Omega} + \dots,$$

$$S \simeq S_0 + \ln\frac{T}{\Delta} + \dots \tag{16}$$

If we take the large system size limit first, we get a non-vanishing zero temperature entropy. Whereas naive extrapolations to $T \to 0$ limit gives vanishing $e^S$ and a finite energy. We refer the reader to the discussion on the Sachdev-Ye-Kitaev model in section 2, where this curious feature is commented on.

3. *Very low temperatures* ($T \ll \Delta/\Omega$): In this regime, $\beta\Delta/\Omega \gg 1$. Hence

$$
\begin{aligned}
\ln Z &= -\frac{\Delta\beta}{\Omega} + e^{-\frac{\Delta\beta}{\Omega}} + \dots, \\
U &\simeq \frac{\Delta}{\Omega} + \frac{\Delta}{\Omega}e^{-\frac{\Delta\beta}{\Omega}} + \dots, \\
S &\simeq \frac{\beta\Delta}{\Omega}e^{-\frac{\beta\Delta}{\Omega}} + \dots
\end{aligned}
\tag{17}
$$

In zero temperature limit, the energy approached that of the true ground state and the entropy vanishes.

The "low but not too low" temperature regime $\Delta \gg T \gg \Delta/\Omega$ is of particular interest. Firstly we note that a similar temperature regime exists for the SYK model[9] as discussed towards the end of section 2. The temperature in SYK model should be low in order to access the deep infrared, but not so low that the exponentially small energy gap over the ground state becomes evident and the true ground state reigns supreme.

This temperature regime also includes the natural temperature[10] associated with the low lying states in the energy interval $(\Delta/\Omega, \Delta)$,

$$
T_{natural}^{-1} := \frac{\Delta S}{\Delta E} \simeq \frac{\ln\Omega}{\Delta}.
\tag{18}
$$

For large $\Omega$, which is the case for big black holes, the condition $\Delta \gg T_{natural} \gg \Delta/\Omega$ is readily satisfied. The derivation of (16) assumes a hierarchy between the energy scales, $\Delta/\Omega$ and $\Delta$. This assumption is clearly justified for large black holes. But even for small charges, where the black hole picture is not trustworthy, $\Omega$ can be large enough to allow for a "low but not too low" temperature regime. E.g. for 1/8 BPS black holes, the smallest ground state degeneracies for smallest charges are $12, 56, 208, 684, \dots$.

This regime's claim to fame is that up to numerical coefficients, (16) has the same behaviour as near-extremal thermodynamics (1) for low temperatures. Comparison of the $\log T$ term in entropies of (16) and (1) reveals that $\Delta \sim M_{gap}$. Note that $\Delta$ is not quite a gap scale. The numerical mismatches are hardly unexpected, since the density of states for near-extremal black holes scale as $\rho(E) \sim \sqrt{E}$ for very small energies (see (2)), whereas we are working with a constant $\rho(E)$ for low energies. It is then remarkable that such a simple picture nevertheless succeeds in capturing the key features. We believe this signifies that the statistical mechanics of near-extremal black holes is essentially that of exponentially many low lying states. In particular, this serves as a useful guideline for modelling near-extremal black holes.

A much debated issue for near-extremal black holes is the presence or absence of a gap. Recent findings seem to suggest absence of a gap [29]. However, the methods used to reach this conclusion are not refined enough to resolve individual microstates. Thus the suggested gapless spectrum might be an approximation for densely packed low lying states. We note that the low energy spectrum considered in this paper is such a spectrum. At low enough energy resolutions, the spectrum at hand would seem gapless (below energy $\Delta$), whereas at higher resolutions, individual microstates are revealed.

Note that equation (16), which entails $e^S \sim T$, does not imply a vanishing $e^S$ at zero temperature, as it is derived for "low but not too low" temperatures. This is unlike the results of [29], which are supposed to be valid even for very low temperatures. One response to this difference could be that the spectrum considered in this paper fails to capture this particular aspect of near-extremal physics. Another possibility, which we would like to consider, is to allow

---

[9]We thank Gustavo Turiaci for discussions on this issue.

[10]Ref. [78] assigned temperature to near-extremal CFT states in a similar fashion.

room for reasonable doubt regarding the applicability of a gravity analysis of near-extremal black holes, to arbitrarily low temperatures.

Lastly, the fact that (16) captures the thermodynamic features of only the Schwarzian part, is significant. The Schwarzian dynamics emanates from the boundary of near$AdS_2$ space [33]. If one deforms the black hole geometry towards extremality, the throat becomes deeper and eventually the near horizon $AdS_2$ decouples. On the $AdS_2$ boundary lives the dual $CFT_1$ which in turn is obtained by letting the microscopic D-brane quantum mechanics flow to the deep infrared, where it captures only the ground states. This suggests that the states of the Schwarzian theory are closely related to the states of the $CFT_1$, i.e. the degenerate ground states of the D-brane quantum mechanics. Recovery of qualitative features of the Schwarzian thermodynamics from a system of near-degenerate low lying states, then seems to suggest that these low lying states correspond to the ground states of the corresponding extremal black hole.

## 4  Discussion

We have shown that up to numerical coefficients, the low energy Schwarzian thermodynamics can be explained by a simple ansatz hypothesizing exponentially many low lying states, separated by a gap from rest of the spectrum. We have argued such spectra are likely to arise in microscopic D-brane descriptions. Interestingly, the scale $M_{gap}$ arises not from a gap in the spectrum, but from the width $\Delta$ of the band of the low lying states. If these low lying states are result of degeneracy lifting, then the width $\Delta$ is naturally associated with the degeneracy lifting scale. Note the same scale appears in $AdS_2$ isometry breaking in black hole side. This in particular suggests that low energy states of the Schwarzian theory are none other than the states of the corresponding extremal black hole, albeit degeneracy lifted, the lifting scale being proportional to the $AdS_2$ isometry breaking scale. Altogether, we get the hint that the low lying states of the Schwarzian theory are none other than low lying states in microscopic description. It seems likely then that the spectrum changes smoothly as one goes from D-brane to black hole description [45].

If the low energy Schwarzian part arises from the near-degenerate low lying states, where does the rest of near-extremal thermodynamics come from? The natural answer seems to be the rest of the spectrum. To be more precise, for temperatures well above the scale $\Delta$, the thermodynamics of the system should be well explained by the excited states above the near-degenerate low lying states. Although explicit descriptions of D-brane quantum mechanics are readily available (e.g. [20,21]), they are likely to be inadequate for checking this, since we do not know how the spectrum is modified as one moves from D-brane regime to the black hole regime.

Another intriguing possibility hinted at by our findings is realization of a $nCFT_1$ as low energy theory of a D-brane system with exponentially many low lying states. For extremal black holes, at least for supersymmetric ones, $AdS_2/CFT_1$ correspondence is realized as follows [79]. The somewhat trivial $CFT_1$ arises as the infrared fixed point of the gapped D-brane quantum mechanics containing exponentially many ground states (e.g. [20,21]), whereas the dual $AdS_2$ arises as the near horizon geometry of the corresponding black hole. For near-extremal black holes, the NHR region leads to a $nAdS_2$ space and one has a $nAdS_2/nCFT_1$ correspondence [33,80]. It is natural to wonder whether the $nCFT_1$ can arise as low energy effective theory of a D-brane quantum mechanics. Our findings embolden this expectation, and suggests such a D-brane quantum mechanics should have exponentially many low lying states. Note, that such a D-brane quantum mechanics is conceivable even for extremal non-supersymmetric states.

Perhaps the most intriguing implication of this work is for strange metals. Strange metals have long been related to black holes [81,82]. For example, the Planckian timescale [46,47], which controls the electronic dynamics in cuprate strange metals [47], resembles the ring down timescale of black holes [48, 49]. One way to realize Planckian transport has been based on models without quasi particles [71,83], or equivalently as systems with exponentially many low lying states. Our work suggests the possibility that these states might be realized as degeneracy lifting of a highly degenerate ground state, with degeneracy protected by some symmetry (possibly emergent). Emergent supersymmetry in disordered systems might be one such symmetry [84].

## Acknowledgments

We would like to thank Gustavo Turiaci for useful discussions.

**Funding information**   This work was supported by Dublin Institute for Advanced Studies and Banaras Hindu University.

## A   Exponentially many low lying states from supersymmetric quantum mechanics

Here we provide an explicit realization of exponentially many low lying states by soft degeneracy lifting of degenerate ground states. Exact degeneracy is a rather fine tuned situation, and is supposed to be destroyed unless protected by some symmetry. Such a symmetry could be supersymmetry, as supersymmetry algebra demands that a supersymmetric ground state must have exactly zero energy. If the system has a sizeable entropy at zero temperature, as is the case for supersymmetric black holes, then the corresponding supersymmetric quantum mechanics (e.g. [20,21]) has exponentially many degenerate ground states. If the supersymmetry is softly broken, the ground state degeneracy would be lifted leading to exponentially many low lying states.

To demonstrate the soft degeneracy lifting, it suffices to consider a bunch of $\mathcal{N} = 1, d = 4$ chiral multiplets, captured by a Lagrangian

$$L = \partial_\mu \phi_i^\dagger \partial^\mu \phi_i - i\bar{\psi}_i \bar{\sigma}^\mu \partial_\mu \psi_i - \frac{1}{2} \frac{\partial^2 W}{\partial \phi_i \partial \phi_j} \psi_i \psi_j - \frac{1}{2} \frac{\partial^2 \overline{W}}{\partial \phi_i^\dagger \partial \phi_j^\dagger} \bar{\psi}_i \bar{\psi}_j - \left(\frac{\partial W}{\partial \phi_i}\right)\left(\frac{\partial W}{\partial \phi_i}\right)^\dagger . \quad \text{(A.1)}$$

Here $(\phi_i, \psi_i)$ respectively are the complex scalar and Weyl spinor in $i^{th}$ chiral multiplet, and $W(\phi)$ is the superpotential.

Let $\phi_0 := \{\phi_{0,i}\}$ be an extremum of the superpotential and hence a classical supersymmetric vacuum of the theory. Around the extremum, we can expand $\phi_i = \phi_{0,i} + \varphi_i$. The zero mode dynamics around any extremum is captured by the Lagrangian with disjoint bosonic and fermionic dynamics

$$L = \dot{\varphi}_i^\dagger \dot{\varphi}_i - i\bar{\psi}_i \dot{\psi}_i - \frac{1}{2} M_{ij} \psi_i \psi_j - \frac{1}{2} \overline{M}_{ij} \bar{\psi}_i \bar{\psi}_j - M_{ij} \overline{M}_{ik} \varphi_j \overline{\varphi}_k + \dots , \quad \text{(A.2)}$$

where $M_{ij} = \left. \frac{\partial^2 W}{\partial \phi_i \partial \phi_j} \right|_{\phi = \phi_0}$. This is same as the system discussed in subsection (3.1), with $M' = 0$.

Existence of multiple classical minima then corresponds to multiple quantum ground states with zero energy. A soft explicit breaking of supersymmetry would lift this degeneracy, yielding many closely spaced low lying states. In the following we demonstrate the essential point for a single chiral multiplet. A simple way to explicitly break supersymmetry would be to add a small bosonic potential, without any fermionic counterpart. Perhaps the simplest choice for the modified potential would be

$$V_B = \left| \frac{\partial W}{\partial \phi} \right|^2 + \epsilon |\phi|^2 \,, \tag{A.3}$$

with $\epsilon^2 \ll \left. \frac{\partial^2 W}{\partial \phi^2} \right|_{\phi=\phi_0}$, for all extrema $\phi_0$ of the superpotential $W$. For multiple chiral fields, $\epsilon^2$ should be much smaller than the smallest eigenvalue of the Hessian $\partial_i \partial_j W$.

Now we determine the change in low energy dynamics due to $\epsilon |\phi|^2$ piece. The change is two fold, first the change in location of the potential minima (which would generically have non-zero potential) and change in the frequencies of the harmonic oscillators. Given an extremum $\phi_0$ of $W$, let the shifted minima of $V_B$ be at $\phi_0 + \varphi_0$. Expanding $\phi = \phi_0 + \varphi_0 + \rho$, around $\phi = \phi_0$ we have

$$\begin{aligned}
V_B &\simeq |M|^2 |\varphi_0 + \rho|^2 + \epsilon |\phi_0 + \varphi_0 + \rho|^2 \\
&= |M|^2 |\varphi_0|^2 + \epsilon |\phi_0 + \varphi_0|^2 + \left( \epsilon \phi_0^* + (|M|^2 + \epsilon) \varphi_0^* \right) \rho \\
&\quad + \left( \epsilon \phi_0 + (|M|^2 + \epsilon) \varphi_0 \right) \rho^* + (|M|^2 + \epsilon) |\rho|^2 \,.
\end{aligned} \tag{A.4}$$

Demanding $\rho = 0$ to be a minima of $V_B$, we find $\varphi_0 = -\epsilon \frac{\phi_0}{|M|^2 + \epsilon}$. Note the value of the potential at the minima is now

$$V_0 = |M|^2 |\varphi_0|^2 + \epsilon |\phi_0 + \varphi_0|^2 = \epsilon \frac{|\phi_0|^2}{|M|^4} + \dots \tag{A.5}$$

Since different minima will have different $\phi_0, M$, they are non-degenerate. Perhaps a better description would be near-degenerate as the difference of energy among various minima is of $\mathcal{O}(\epsilon)$, which we have taken to be very small.

Even the quantum ground state energies no more vanish and differ for different minima. The bosonic oscillators now have frequencies $\approx |M| + \frac{\epsilon}{2|M|}$. For the fermionic oscillators, $\left. \frac{\partial^2 W}{\partial \phi^2} \right|_{\phi=\phi_0}$ will be replaced by

$$\left. \frac{\partial^2 W}{\partial \phi^2} \right|_{\phi=\phi_0 + \varphi_0} = M + \varphi_0 \left. \frac{\partial^3 W}{\partial \phi^3} \right|_{\phi=\phi_0} + \dots \,, \tag{A.6}$$

which depends on the detail of $W$.

Anyhow, the bosonic and fermionic frequencies are different, thus the the ground state energy is no more zero but some $\mathcal{O}(\hbar \epsilon)$ number. We have reinstated $\hbar$ to stress that this quantum ground state energy is on top of a classical ground state energy $V_0$ (A.5).

Since the modified ground state energies depend on the detail of the particular extrema of the superpotential, ground states corresponding to different minima would be different. But the difference will be of $\mathcal{O}(\epsilon)$, hence for small $\epsilon$, such difference will be small as well, leading to large number of closely spaced low lying states, one corresponding to each minimum of the superpotential. Taking perturbative corrections as well as tunnelling effects, will change the detail. But the broad picture, i.e. large number of densely packed low lying states will hold true.

Note that excited states are separated by a gap of $\mathcal{O}(\hbar|M|)$, we would like $V_0$ to lie well below this gap scale. While it is somewhat odd to demand a classical potential minima to be small by quantum standards, there is nothing inconsistent at a formal level. E.g. the classical potential in a supersymmetric quantum mechanics has $\hbar$-dependent terms. In an actual physical problem, the supersymmetry breaking term itself might have quantum origins.

## B  Average level spacing

**Spectrum with two states:**  Consider 2 points randomly drawn from uniform probability density distribution over the interval $[0, 1]$. The probability of them being distance $l$ apart is

$$\pi(l) = \int_0^1 (p_1(x)p_2(x+l) + (1 \leftrightarrow 2))\,dx = 2(1-l)\,, \tag{B.1}$$

where $p_i(x)dx$ is the probability of $i^{th}$ particle being in the interval (x, x+dx). For uniform probability $p_i(x) = 1$ for $x \in [0, 1]$ and 0 elsewhere. Note $\int_0^1 \pi(l)dl = 1$. The average separation is then

$$\bar{l} = \int_0^1 l\pi(l)\,dl = 1/3\,. \tag{B.2}$$

**Spectrum with three states:**  Consider 3 points randomly drawn from uniform probability density distribution over the interval $[0, 1]$. The probability of them being consecutively at distance $l_1, l_2$ apart is

$$\pi(l_1, l_2) = \int_0^1 p_1(x)p_2(x+l_1)p_3(x+l_1+l_2)\,dx + \text{permutations} = 6(1 - l_1 - l_2)\,. \tag{B.3}$$

Note $l_1, l_2$ can are constrained by $l_1 + l_2 \leq 1$. With this in mind $\pi(l_1, l_2)$ is seen to have the correct normalization: $\int_0^1 dL_2 \int_0^{L_2} \pi(l_1, l_2)dl_1 = 1$, where $L_2 = l_1 + l_2$. The average value of $l_1$ then is

$$\overline{l_2} = \overline{l_1} = 6\int_0^1 dL_2 \int_0^{L_2}(1-L_2)l_1 dl_1 = 6\int_0^1 dL_2\left(\frac{L_2^2}{2} - \frac{L_2^3}{2}\right) = 1 - \frac{3}{4} = 1/4. \tag{B.4}$$

**Spectrum with four states:**  Similarly for 4 randomly chosen points, one finds

$$\pi(l_1, l_2, l_3) = 4!(1 - l_1 - l_2 - l_3)\,,$$
$$\overline{l_3} = \overline{l_2} = \overline{l_1} = \int_0^1 \int_0^{L_3} \int_0^{L_2} \pi(l_1, l_2, l_3)l_1\,dl_1 dL_2 dL_3 = 1/5\,.$$

**Spectrum with $\Omega$ states:**  On similar lines, it can be shown that for $\Omega$ randomly chosen points

$$\pi(l_1, l_2, \ldots, l_{\Omega-1}) = \Omega!(1 - l_1 - l_2 - \cdots - l_{\Omega-1})\,,$$
$$\bar{l}_1 = \bar{l}_2 = \cdots = \bar{l}_{\Omega-1} = 1/(\Omega+1)\,. \tag{B.5}$$

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
