# Peer review of "Statistical Mechanics of Exponentially Many Low Lying States"

_SciPost Physics, doi:SciPost Phys. 18, 103 (2025)_

## Round 2 · Referee Report · Anonymous (Referee 1) · 2024-9-18

Report

The subject of the paper is well-aligned with the journal's theme. However, before accepting for publication, some changes need to be incorporated. This is mentioned in detail in the report attached.

Attachment

Recommendation

Ask for major revision

  • validity: -
  • significance: -
  • originality: -
  • clarity: -
  • formatting: -
  • grammar: -

Author:  Swapnamay Mondal  on 2025-01-01  [id 5074]

(in reply to Report 1 on 2024-09-18)

We thank the referee for reviewing the manuscript and providing valuable comments. In light of the referees’ comments, we have made necessary changes in the manuscript. in the following, we address the referee's comments point by point.

  1. Reply to referee's comment 1:

We thank the referee for pointing this out. We have changed that specific sentence (first sentence of last paragraph in page 3) accordingly.

  1. Reply to referee's comment 2:

Many thanks for this feedback. We have completely rewritten the part on SYK model, expanding significantly on the last version.

  1. Reply to referee's comment 3:

We have corrected this in the revised manuscript.

  1. Reply to referee's comment 4:

The relevant brane system has not been discussed in literature to the best of our knowledge, except my recent very recent work “An extremal black hole with a unique ground state” (https://arxiv.org/abs/2411.11096). I was not sure if it would be appropriate to cite a future paper, so did not mention it in the manuscript.

  1. Reply to referee's comment 5:

As for the first question, the energies of each such “ground state” depends on the specifics of the corresponding classical minima. Since different minima will generically different specifics, (e.g. different M) corresponding ground state energies will also be different. If there is some discreet symmetry, then the minima related by this symmetry will have perturbatively degenerate ground states, which would nevertheless be lifted by non-perturbative effects. We have added a footnote in page 9 explaining this.

Coming to the second question, we suppose the referee’s concern is about the now non-zero ground state energies, since the classical minima have been taken to be close enough by taking the supersymmetry breaking scale \epsilon to be very small. Note that ground state energies are all zero when \epsilon=0, thus by continuity can be made very small, by choosing \epsilon accordingly. In particular we can choose them to be well below the first excited state in the supersymmetric quantum mechanics. That way we will have exponentially many low lying states. We have added this explanation in last two paragraphs of Appendix A.

  1. Reply to referee's comment 6:

We thank the referee for this question. We did not succeed in proving that a generic spectrum is nearly equi-spaced. But we could prove a somewhat weaker statement- if a fixed number of states are distributed randomly over a fixed interval, then the average spacing between any two consecutive states is that of the equi-spaced spectrum. So the equi-spaced spectrum captures the average situation. 

We have relegated the proof in the appendix B and re-written initial parts of section 3.2 in light of this result. In particular, we do not claim a generic spectrum to be featureless and hence well approximated by an equi-spaced one.

  1. Reply to referee's comment 7:

We agree with the referee that E_{gap} does not seem to have a gravity analog. But we think this is not worrying for the following reasons. 

The limited concern of our manuscript is the low temperature thermodynamics, which will be dominated by the low lying states anyway. So even if there is no E_{gap}, but only comparatively sparse states above the band of low lying states, we do not expect the low temperature thermodynamics to change significantly. Thus for the purpose of this paper, E_{gap} should be treated as a simplifying assumption, which need not have a counterpart in gravity.

We would also like to mention that gravity analysis itself should be treated as an approximation to the true microscopic situation. E.g. the gravity computations do not make the discreet nature of the spectrum apparent. Thus the real question is whether relevant D-brane descriptions exhibit an E_{gap} or not, which is beyond the scope of this paper.

---

## Round 2 · Referee Report · Anonymous (Referee 2) · 2024-10-18

Report

This paper studies the low-temperature corrections to entropy and energy by analyzing the statistical mechanics of low-energy states. These corrections are calculated in the context of a uniform band of low-energy states that are separated from the high-energy states by a large gap.

I recommend the paper for publication after minor changes that are explained below.

(1) Section 3.2 tries to argue that the partition function of a generic spectrum, $Z_{generic}$, is well-approximated by choosing the energy levels to be equally spaced, $Z_{equi-spaced}$.

This is done numerically by comparing $Z_{generic}$ and $Z_{equi-spaced}$. However, the numerical analysis presented in the paper is not sufficient when the temperature is small. The paper chooses $\Delta = 1$ and the plots in Figures 1-5 show $T \in [0,1]$. However, most of the relative error plots showing $(Z_{generic} - Z_{equi-spaced})/Z_{equi-spaced}$ tends to blow up for small $T$. This is problematic because the paper is very interested in the temperature range $T \ll \Delta$. More evidence is needed to show that $Z_{generic} \approx Z_{equi-spaced}$ in this temperature range.

(2) Relatedly, it is not clear why $Z_{generic}$ obtained from taking the energy levels to be independent, uniformly distributed random variables is a good ansatz for the cases under consideration.

For instance, if we assume that the energy levels are given by a matrix model then we expect the density of states to have a square root edge, i.e. $\rho(E) \sim \sqrt{E}.$ Moreover, the energy levels are not independent but they level repel. This does not match with the assumptions for $Z_{generic}$. Some comments on this would be helpful.

(3) In section 3.3, case 2 studies the low but not too low temperatures. Some passing comments are made below equation (3.14) but these should be explained better. It would be helpful if the large system limit is explained better. Also, it is not clear why $T \to 0$ is a valid limit in this case.

Recommendation

Ask for minor revision

  • validity: good
  • significance: ok
  • originality: good
  • clarity: high
  • formatting: good
  • grammar: excellent

Author:  Swapnamay Mondal  on 2025-01-01  [id 5073]

(in reply to Report 2 on 2024-10-18)

We thank the referee for reviewing our manuscript and for his/her comments. In the following we address the referee’s comments point-wise.

  1. Reply to the first comment:

We are not really interested in very low temperatures, but in ``low but not too low temperatures” \Delta/\Omega << T << \Delta. E.g. in figure 3, \Delta=1, \Delta/\Omega=.001 . For intermediate temperatures around .1, the maximum value of the relative error ranges between 0.2% to 6%, which is not large.

Anyhow we have rewritten this section. In particular, we show (in appendix B of the revised manuscript) that should one draw the individual eigenvalues at random from a uniform distribution, then the statistical average of spectra thus generated is indeed the equi-spaced spectrum. Although this by itself does not prove the equi-spaced spectrum to be generic, along with numerical evidences, which remain unchanged, we believe for the limited purpose of our paper (i.e. partition function in a specific temperature regime) this is a reasonable approximation.

  1. Reply to the second comment:

We would like to clarify that we are not claiming to reproduce the exact low temperature near-extremal thermodynamics. Our limited concern is - what sort of discreet spectra captures the qualitative features and we explicitly show the equi-spaced spectrum indeed does. Yes, it is clearly different from near-extremal black holes or matrix models, as pointed out by the referee. 

As for the question why this should be a good ansatz to start with, all we know (as discussed in the manuscript) about such brane systems is that they are very are likely to have exponentially many low lying states. With this limited information, we looked at the simplest solvable case and that happened capture the qualitative features. Surely it is conceivable that one might consider a different spectrum that is closer to the physical systems considered. 

Anyhow, our a posteriori understanding is that the log T piece in entropy is not specific to near-extremal black holes or Schwarzian theory, but to a wider class of systems, with the common feature being the presence of exponentially Manny low lying states.

  1. Reply to the third comment:

We thank the referee for this comment. In the revised manuscript we have explained this subtle issue in some detail in the part of section 2 describing Sachdev-Ye-Kitaev model. Below equation 3.14 we have referred to discussions in this Sachdev-Ye-Kitaev model section.

---

## Round 2 · Referee Report · Anonymous (Referee 3) · 2024-10-22

Strengths

1. well motivated
2. well explained
3. important results
4. detailed numerical analysis

Weaknesses

1. comparison with existing results
2. physical reasoning for temperature cutoff

Report

recommended for publication

Requested changes

1. Physical reasoning for temperature cutoff

Recommendation

Publish (easily meets expectations and criteria for this Journal; among top 50%)

  • validity: high
  • significance: good
  • originality: high
  • clarity: high
  • formatting: excellent
  • grammar: good

Author:  Swapnamay Mondal  on 2025-01-01  [id 5072]

(in reply to Report 3 on 2024-10-22)

We thank the referee for finding our article worthy of publication in SciPost Physics as well as for his comments.

We suppose, by temperature cutoff the referee refers to our suggestion that the gravity analysis is not valid below a temperature. This is related to the “low but not too low” temperature regime advocated in our paper, which is discussed in some detail in “Sachdev-Ye-Kitaev model” part of section 2 of the revised manuscript. This discussion is also referred to below equation 3.14.

---

## Round 3 · Referee Report · Anonymous (Referee 1) · 2025-1-10

Report

The author has addressed my queries and comments in detail and made necessary changes to the manuscript.

Recommendation

Publish (easily meets expectations and criteria for this Journal; among top 50%)

---

## Round 3 · Referee Report · Anonymous (Referee 2) · 2025-2-1

Report

The author has made the requested changes and resolved the issues raised in my previous report. I recommend this paper for publication.

Recommendation

Publish (meets expectations and criteria for this Journal)

---

## Round 3 · List of Changes

1. In the second paragraph of the introduction section, we have added few lines mentioning that negative infinite entropy at zero temperature can not be an exact statement for a discreet spectrum.

2. In the paragraph before equation 2.1, we have replaced the first sentence “Slow variation of transverse S2 in NHR breaks the AdS2 isometries” by “Both the AdS2 and S2 receive temperature corrections of same order, leading to the breaking of AdS2 isometries in the NHR.” 

3. The part of section 2, dealing with SYK model is expanded.

4. Initial parts of section 3.2 is completely rewritten.

5. Appendix B is added, where it is shown that the “average spectrum” with individual states drawn randomly from uniform distribution, is the equi-spaced distribution considered by us.

---

## Editorial Decision

published